# Investigating the HIV epidemic among Black gay and bisexual men in the Southern United States: Results of the HPTN 096 pilot cross-sectional assessment

Chris Beyrer[1]*, Robert H. Remien[2], Susan H. Eshleman[3], Theresa R. Gamble[4], Jean De Dieu Tapsoba[5], Rita L. Labbett[4], Philip A. Sullivan[3], Oliver Laeyendecker[6], Peter L. Anderson[7], Devang Agravat[5], James P Hughes[5], Daniel D. Driffin[8,9], Craig S. Hutchinson[10], Christopher Hucks-Ortiz[10,11], Maurice Adair[12], Melissa Curry[13], Shakita Brooks Jones[14], Ian L. Haddock[15], Donte Boyd[16], Dale R. Burwen[17,18], Anna Satcher Johnson[19], LaRon E. Nelson[20,21,22] on behalf of the HPTN 096 pilot study team¶

1 Duke Global Health Institute, Duke University, Durham, North Carolina, United States of America, 2 New York State Psychiatric Institute and Columbia University, New York, New York, United States of America, 3 Johns Hopkins University School of Medicine, Baltimore, Maryland, United States of America, 4 HIV Prevention Trials Network Leadership and Operations Center, Durham, North Carolina, United States of America, 5 Vaccine and Infectious Disease Division, Fred Hutchinson Cancer Center, Seattle, Washington, United States of America, 6 Division of Intramural Research, National Institute of Allergy and Infectious Diseases, National Institutes of Health, Bethesda, Maryland, United States of America, 7 Department of Pharmaceutical Sciences, Skaggs School of Pharmacy and Pharmaceutical Sciences. University of Colorado Anschutz Medical Campus, Aurora, Colorado, United States of America, 8 HIV Prevention Trials Network 096 Community Advisory Board, Atlanta, Georgia, United States of America, 9 D3 Consulting Limited Liability Company, Atlanta, Georgia, United States of America, 10 HIV Prevention Trials Network Black Caucus, Columbia, Washington, United States of America, 11 Equity Solutions, Limited Liability Company, Columbia, Washington, United States of America, 12 AID Upstate, Greenville, South Carolina, United States of America, 13 Abounding Prosperity, Dallas, Texas, United States of America, 14 Central Alabama Alliance, Resource & Advocacy Center, Montgomery, Alabama, United States of America, 15 The Normal Anomaly, Houston, Texas, United States of America, 16 The Ohio State University, College of Social Work, Columbus, Ohio, United States of America, 17 Affiliation at the time the study was conducted: Division of AIDS, National Institute of Allergy and Infectious Diseases, National Institutes of Health, Rockville, Maryland, United States of America, 18 Current affiliation: Division of Clinical Innovation, National Center for Advancing Translational Sciences, National Institutes of Health, Rockville, Maryland, United States of America, 19 Division of HIV Prevention, National Center for HIV, Viral Hepatitis, Sexually Transmitted Disease and Tuberculosis Prevention, Centers for Disease Control and Prevention, Atlanta, Georgia, United States of America, 20 School of Nursing, Yale University, New Haven, Connecticut, United States of America, 21 Department of Social and Behavioral Sciences, School of Public Health, Yale University, New Haven, Connecticut, United States of America, 22 Center for Interdisciplinary Research on AIDS, School of Public Health, Yale University, New Haven, Connecticut, United States of America

¶ Membership of the HPTN 096 pilot study team is provided in the Acknowledgements
* chris.beyrer@duke.edu

## Abstract

### Background

The HIV Prevention Trials Network (HPTN) 096 study was designed to address the markedly higher rates of HIV incidence among Black men who have sex with men (MSM) in the Southern United States (US). A cross-sectional assessment was

---

**Data availability statement:** The data supporting the findings of this study are available at the Harvard Dataverse at https://dataverse.harvard.edu/dataverse/HPTN096/.'

**Funding:** Overall support for the HIV Prevention Trials Network (HPTN) is provided by the National Institute of Allergy and Infectious Diseases (NIAID), Office of the Director (OD), National Institutes of Health (NIH), National Institute on Drug Abuse (NIDA), the National Institute of Mental Health (NIMH), and the Eunice Kennedy Shriver National Institute of Child Health and Human Development (NICHD) under Award Numbers UM1AI068619 (HPTN Leadership and Operations Center), UM1AI068617 (HPTN Statistical and Data Management Center), and UM1AI068613 (HPTN Laboratory Center.

**Competing interests:** Both DRB and MJS served as representatives of the funder, NIAID and NIMH, respectively, and both played roles in the study design, data analysis and interpretation, decision to publish, and preparation of the manuscript. Both DDD and CHO received funding from NIH through the award identified in the funding statement for their participation in study design, data analysis and interpretation, and preparation of the manuscript; however, their individual LLCs (D3 Consulting and Equity Solutions) did not play any role in these activities. These competing interests do not alter our adherence to PLOS ONE policies on sharing data and materials.

conducted during the pilot phase of the study to determine its feasibility and collect key HIV-related metrics for the study population.

## Methods and findings

Four hundred and twenty-two Black MSM, ≥ 15 years old and living in the four pilot communities (Dallas, TX; Houston, TX; Montgomery, AL; Greenville, SC), were enrolled via starfish sampling into the cross-sectional assessment. Each participant completed two questionnaires and had blood samples collected at a single study visit. Laboratory testing was performed to determine HIV status and use of oral pre-exposure prophylaxis (PrEP). HIV drug resistance and viral suppression were also assessed for two of the four pilot communities (Dallas and Houston). Categorical variables were summarized using frequency and percentage. Continuous variables were summarized using mean, standard deviation, median and interquartile range. Univariable and multivariable logistic regression models were used to assess various associations. HIV status was determined for 403 of the 422 participants (95.5%); 212 (52.6%) men were living with HIV, including one with acute HIV. For these participants, 163 (76.9%) reported that they were in HIV care. In Dallas and Houston, 71 of the 101 living with HIV (70.3%) were virally suppressed. Of the 191 not living with HIV, 57 (29.8%) reported ever taking PrEP, 41 (21.5%) reported being currently on PrEP, and eight (4.2%) reported never having heard of PrEP. PrEP use was documented through laboratory testing in 36 (19.1%) of 188 participants tested; of the 41 participants reporting current PrEP use, five did not have laboratory evidence of PrEP use.

## Conclusion

During the pilot, we successfully recruited Black MSM using starfish sampling and demonstrated the feasibility of collecting primary study outcomes using a cross-sectional assessment. We found a high burden of HIV and those living with HIV had only a moderate rate of viral suppression. In addition, PrEP use was uncommon among the men living without HIV. Reducing HIV incidence in Black MSM remains a key element to addressing the HIV epidemic in the US.

## Introduction

The recent Centers for Disease Control and Prevention (CDC) report on HIV diagnosis in the United States (US) underscores the ongoing health disparity in HIV acquisition among men who have sex with men (MSM), with Black and Latino MSM, as well as MSM living in the Southern US, bearing the highest burden of infection [1]. Though decreasing, HIV incidence and prevalence are markedly higher among Black MSM when compared to their White counterparts. This disparity has persisted over two decades despite increased HIV testing and advances in the use of antiretroviral

drugs for both HIV treatment and prevention [2,3]. Multiple studies have shown that the high rates of HIV acquisition among these men cannot be ascribed to individual characteristics alone, but are significantly attributable to network, social and structural factors—the social determinants of health [4].

HIV Prevention Trials Network (HPTN) 096 was initially designed as a controlled, community-randomized, type-two hybrid efficacy/implementation study to evaluate the impact of an HIV status neutral intervention designed to address the social determinants of HIV outcomes among Black MSM in the Southern US. Specifically, the study aimed to increase the proportion of Black MSM living with HIV who are virally suppressed and to increase the proportion of Black MSM with risk factors for HIV acquisition who initiate pre-exposure prophylaxis (PrEP). The intervention itself was a four-component integrated strategy that includes a healthcare facility-based intersectional stigma reduction intervention; a peer support program; the use of social media; and a health equity intervention based on local coalition building and supporting collective community actions to address social determinants of health.

In 2022, we conducted a pilot to determine the feasibility of both the intervention and the cross-sectional assessment, which was how the primary study outcomes would be measured. In addition to determining feasibility, the pilot cross-sectional assessment was used to collect key metrics, including HIV prevalence, viral suppression and PrEP use. The cross-sectional assessment was conducted in four Southern US cities (Dallas and Houston, TX; Montgomery, AL: Greenville, SC) using starfish sampling to recruit Black MSM. Starfish sampling is a hybrid approach that combines elements of venue time and respondent driven sampling to enhance the recruitment of populations when their denominators are unknown, and traditional methods may not work [5]. Only the results of the pilot cross-sectional assessment are reported here.

## Materials and methods

### Study community selection

The initial protocol for the HPTN 096 study included 16 communities in the Southern US, which were selected and matched using the following criteria:

- Geographically distinct (at least 120 miles apart from one another)
- Inclusive of proximal counties where Black MSM frequently travel
- Have roughly equivalent populations (overall, male, Black male)
- Have roughly equivalent HIV prevalence for Black men (target +/- 35%)
- Share similar rates of viral suppression for Black MSM (target +/- 30%)

Population and HIV prevalence parameters were based on data collected for Black men (as opposed to Black MSM) because it was difficult to obtain accurate population estimates for MSM [6]. All 16 study communities selected had a high HIV burden as defined by the US Ending the HIV Epidemic (EHE) initiative [7]. Population data came from the 2019 US census (data.census.gov); HIV prevalence and viral suppression data came from the 2019 CDC HIV surveillance system (www.cdc.gov/nchhstp/atlas/). From these 16 communities, two matched pairs (a total of four study communities) were chosen to participate in the pilot study.

### Eligibility and procedures

The goal of the pilot cross-sectional assessment was to enroll 100 Black MSM in each pilot study community, for a total of 400 men. The enrollment of 100 Black MSM in each pilot community is reflective of the baseline recruitment target of 100 Black MSM per study community for the full study. Individuals were eligible if they were a Black man, reported a lifetime history of anal sex with other men (meaning they had had sex at least once in their life with another man), were at least

15 years of age, reported current residence in one of the pilot study communities, and were willing and able to provide informed consent (or assent if <18 years old), complete two questionnaires and provide blood samples.

The cross-sectional assessment included a short eligibility screen, informed consent, collection of a blood sample, and administration of two surveys (one short, one long). The assessment was incentivized, with participants receiving $50 for blood collection and completing the short survey, and an additional $50 for completing the long survey.

## Sampling methodology

Several approaches have been developed to sample populations for which denominators are not available and for whom true population-based sampling methodologies are challenging or impossible. These include venue time sampling [8] and respondent-driven sampling (RDS) [9]. Starfish sampling is a recent innovation that uses components of each of these established methodologies to enhance recruitment and sampling validity among communities and populations for which we do not have denominators [5]. Starfish sampling was chosen for the cross-sectional assessment because it is an innovative way to sample populations for which no known sampling frame exists.

To implement the first component of starfish sampling, each recruitment team identified venues where Black MSM could be approached and invited to participate in the study. Venues could include specific locations (e.g., gay clubs or bookstores), community events (e.g., Juneteenth or house ball events) and events that the recruitment teams hosted themselves. To preserve the methodology of semi-random venue time sampling, the recruitment teams pre-determined who to approach for recruitment depending on the size of the event. The recruitment teams usually approached every tenth person who passed their table/booth at large events (>200 attendees), every fifth person at medium events (between 100–200 attendees) and every third person at small events (<100 attendees). This allowed flexibility based on event size and ensured that a semi-random sample of men were recruited for a more robust and representative sample of the community.

To implement RDS recruitment, the second component of starfish sampling, everyone who enrolled in the cross-sectional assessment, regardless of how they were recruited, was offered three coupons to give to other Black MSM in their social networks. These coupons served as an invitation to participate in the cross-sectional assessment; when someone with a coupon was enrolled, the person who gave them the coupon received an incentive ($10 per coupon, up to $30 total). The coupons had an expiration date of approximately two weeks to encourage quick coupon return; however, recruitment teams were instructed to accept coupons past their expiration date until the end of enrollment. In addition, staff reached out to participants who received coupons for distribution to remind them to give away their coupons or to encourage those to whom they had given coupons to contact the recruitment team for screening and potential enrollment.

## Survey content and administration

We used standard methods to develop both the short and long surveys. The short survey collected key information on population demographics; awareness of the study; HIV testing and status; prevention and care engagement; and PrEP knowledge and use. The long survey collected supplemental information on social support; intersectional stigma; connection to their community; barriers to HIV testing, engagement in HIV prevention or care services and medication (PrEP and antiretroviral therapy [ART]) use and adherence; and screeners for substance use, anxiety, depression and post-traumatic stress disorder. The parameters used in the analysis presented in this manuscript include: population demographics (short survey); HIV testing and status (short survey); prevention and care engagement (short survey); PrEP knowledge and use (short and long surveys); barriers to engagement in HIV prevention and care services (long survey); barriers to use of PrEP/ART (long survey). Both questionnaires were collected as audio, computer-assisted self-interviews (ACASI) via a tablet or phone. The short survey was administered at enrollment and participants could complete the long survey at the same time, or, later, at their convenience.

## Laboratory methodology

Laboratories that support the HPTN are required to use the Lab Data Monitoring System (LDMS) (https://www.ldms.org/). The Montgomery and Greenville communities did not have an LDMS laboratory that was close enough to allow sample transport, processing, and plasma storage within the required time windows for viral load and HIV drug testing. Thus, plasma samples were only shipped to the HPTN Laboratory Center for viral load and HIV drug resistance testing from Dallas and Houston. Dried blood spot (DBS) from all locations were shipped to the HPTN Laboratory Center for pharmacology testing.

## HIV and viral load testing

Samples collected in all four communities were tested in real-time at the Quest Diagnostics Nichols Institute (San Juan Capistrano, CA). All samples were screened for HIV using the Abbott Architect HIV Ag/Ab Combo Immunoassay (Abbott Diagnostics, Abbott Park, IL); reactive samples were reflexively tested with the Geenius™ HIV 1/2 Supplemental Assay (Bio-Rad Laboratories, Hercules, CA). In addition, all samples were tested using the cobas® HIV-1/HIV-2 Qualitative assay (analytic sensitivity: 12.8 copies/mL for group M HIV-1; Roche Diagnostics, Indianapolis, IN). Plasma was stored for participants enrolled in Dallas and Houston. Those samples were tested at the HPTN Laboratory Center (Johns Hopkins University, Baltimore, MD) to determine HIV viral load using the Panther Aptima HIV-1 Quant Dx assay (Hologic, Marlborough, MA). Participants were classified as having acute HIV infection if HIV RNA was detected with a negative or indeterminate Geenius HIV 1/2 Supplemental Assay result.

## HIV drug resistance testing

Stored samples from participants from Dallas and Houston who had HIV viral loads >500 copies/mL were analyzed using the GenoSure PRIme HIV-1 Drug Resistance assay (Monogram Biosciences, South San Francisco, CA). Drug resistance mutations were identified and predicted drug resistance was determined using the Stanford HIV resistance database v9.4.1 [https://hivdb.stanford.edu] [10].

## Tenofovir-diphosphate testing

Intraerythrocytic tenofovir-diphosphate (TFV-DP) was quantified from DBS using a validated liquid chromatography tandem mass spectrometry method, as previously described [11,12]. Briefly, a 3 mm punch was extracted from DBS arising from unknown regimens or those containing tenofovir disoproxil fumarate (TDF); whereas two 7 mm punches were extracted from DBS arising from regimens containing tenofovir alafenamide (TAF). An updated extraction procedure was used for TFV-DP arising from TDF resulting in slightly higher recoveries [11]. The lower limit of quantification was 31.3 fmol/punch(es).

TFV-DP was categorized as average dosing in the preceding 6–8 weeks, as follows. For unknown and TDF regimens, below quantifiable limit (BLQ) (little to no dosing), BLQ to 399 fmol/punch (< 2 doses per week on average), 400–799 (2–3 doses per week on average) and ≥800 (4 or more doses per week on average). For TAF, TFV-DP categorizations were BLQ (little to no dosing), BLQ to 449 (< 2 doses per week on average, 450–949 (2–3 doses per week on average), and ≥950 (4 or more doses per week on average).

## Statistical methodology

**Outcomes.** The main outcomes of interest were the proportion of participants determined to be living with HIV based on laboratory test results; the proportion of participants living with HIV who were virally suppressed; and the proportion of participants not living with HIV whose laboratory results were consistent with current PrEP use. Participants living with HIV were classified as virally suppressed if the result of the HIV RNA qualitative-based test was "undetected" or the result

of Aptima HIV-1 Quant Dx assay was "not detected" or < 200 copies/mL. Current PrEP use based on DBS testing was defined as having quantifiable TFV-DP. A secondary outcome was the proportion of participants who reported ever using PrEP among participants classified as not living with HIV.

### Statistical analysis

Categorical variables were summarized using frequency and percentage whereas mean and standard deviation (SD), median and interquartile range (IQR) were used to summarize continuous variables. Univariable and multivariable logistic regression models were used to assess associations between established HIV status and age (<25 years, 25 years/older) and recruitment method (RDS, venue-based) in all participants as well as the association between PrEP use ever and age among participants not living with HIV. The analyses were based on complete cases assuming a missing at random mechanism for the missing data. A p-value <0.05 was considered statistically significant for a two-sided test. All analyses were performed using SAS version 9.4 (SAS institute, Cary, NC).

### Human subjects protection and clinicaltrials.gov number

All participants provided written informed consent prior to participation in the cross-sectional assessment. The protocol was overseen by a single Institutional Review Board (Advarra), with the protocol, informed consent forms and all participant-facing materials reviewed and approved prior to use. As participants 15 years and older could be enrolled, the study team requested a waiver of parental consent, which was granted by the single IRB because the research did not involve greater than minimal risk (45 CFR 46.404). HPTN 096 is registered on ClinicalTrials.gov with the following identifier: NCT05075967.

### Protocol deviations

There was one protocol deviation that affected 43 participants who were all enrolled in one study community (Montgomery). All participants were required to have real-time HIV testing performed at Quest Diagnostics: the HIV-1/2 Ag/Ab test with reflex to the discriminatory antibody test, and a qualitative HIV-1 RNA test. For these 43 participants, the HIV-1/2 Ag/Ab test with reflex to the discriminatory antibody test was performed, but the qualitative HIV-1 RNA assay was not.

## Results

### Study community selection

For the initial HPTN 096 study, 16 communities were matched into eight pairs and randomized into two study arms, intervention and standard-of-care (S1 Table). From these, two pairs were chosen to participate in the pilot study: Dallas, TX/ Houston, TX and Montgomery AL/Greenville, SC. These pairs were selected because they represented two larger (Dallas/Houston) and two smaller (Montgomery/Greenville) communities, and two with active Black Treatment Advocates Networks (Houston and Montgomery), as an indicator of local Black leadership and organizations that support Black MSM, and two without (Dallas and Greenville).

### Recruitment and enrollment

Four hundred and twenty-two Black MSM were enrolled into the cross-sectional assessment between May 14, 2022 and November 17, 2022, with 327 enrolled via the venue-based component of starfish sampling recruitment and 95 via the RDS component. (Figs 1 and 2). Enrolled participants (Table 1) had a median age of 33, identified as homosexual or gay (72.2%), had a high school diploma (31.5%) or some college education (34.6%), worked full time (52.0%), earned less than $60,000 (84.9%) annually, had health insurance (64.4%) and were married or living with a partner (78.6%). One hundred and eighty-two (44.0%) self-reported living with HIV. Of those who reported that they were not living with HIV or had

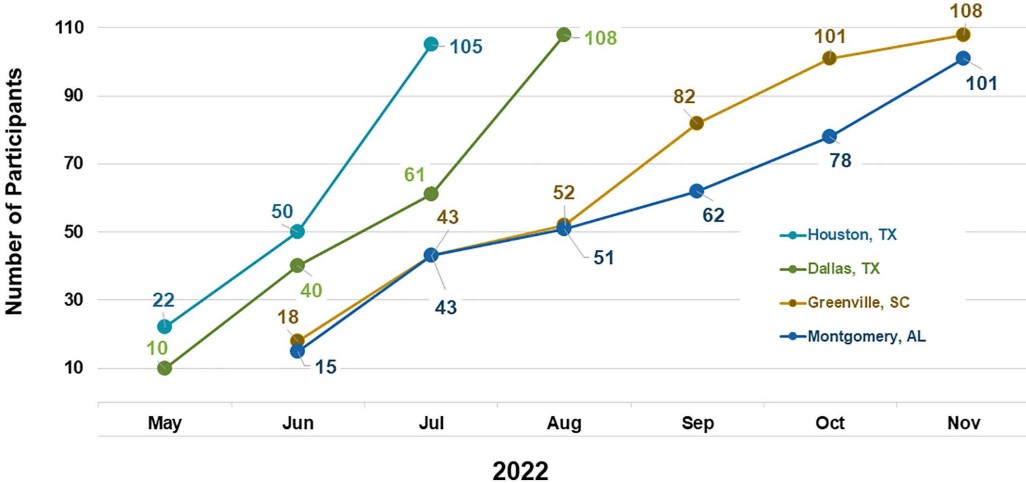

**Fig 1. HPTN 096 CSA Enrollment Results Using Starfish Sampling.**

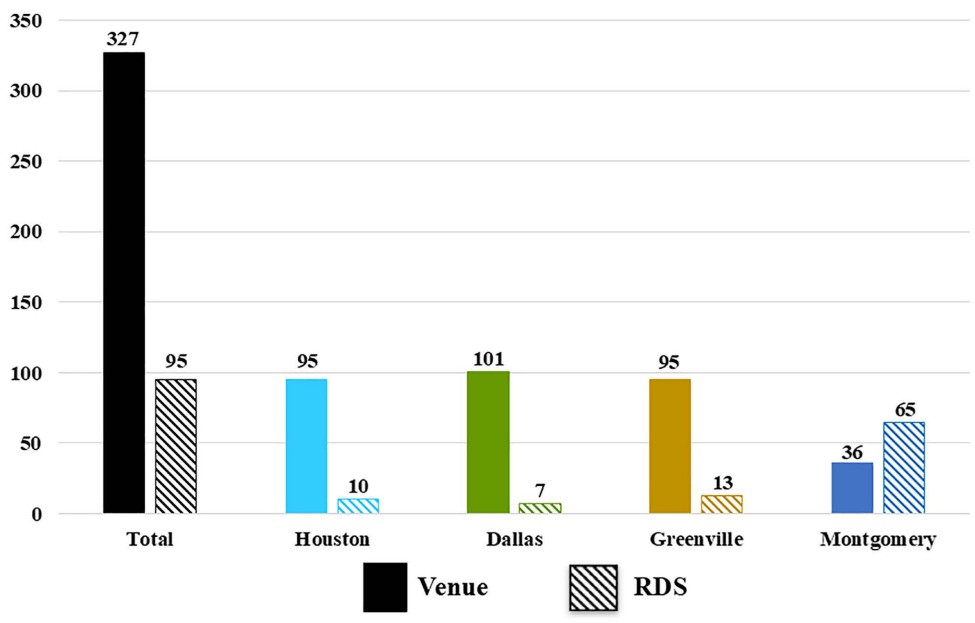

**Fig 2. Enrollment: Venue vs. RDS.**

unknown HIV status, 91.8% reported testing for HIV. Of the 422 enrolled participants, HIV status was determined via laboratory testing for 403 (95.5%). Results were not available for 15 participants because either the participant was not able to provide a blood sample, or because their sample could not be tested due to shipping delays. HIV testing was performed for the remaining four participants without an HIV status determination, but the test results were inconclusive.

There was variation across the study communities regarding the success of the two recruitment methods. Enrollment in the larger study communities (Dallas/Houston) was achieved primarily via venue-based recruitment due to the sheer volume of existing venues and events, with attendance at some events (e.g., Juneteenth events, house balls) including >200

**Table 1. Demographics of Enrolled Participants.**

| Characteristic | Participants |
|---|---|
| **Enrolled** | 422 |
| **Age (years)** | |
| 18−24 | 53/414 (12.8%) |
| 25−34 | 186/414 (44.9%) |
| 35−44 | 86/414 (20.8%) |
| 45−54 | 46/414 (11.1%) |
| Over 55 | 43/414 (10.4%) |
| Missing | 8 |
| Mean Age (SD) | 36.0 (11.5) |
| Median Age (IQR) | 33 (28,42) |
| **Ethnic or Cultural Background** | |
| African American/Black | 401/414 (96.9%) |
| Afro−Latino (e.g., Puerto Rican, Dominican, Cuban, Mexican, etc.) | 14/414 (3.4%) |
| Other, please specify | 15/414 (3.6%) |
| Missing | 8 |
| **Education** | |
| Less than high school | 39/419 (9.3%) |
| High school graduate or equivalent | 132/419 (31.5%) |
| Vocational/trade/technical school | 15/419 (3.6%) |
| Some college | 145/419 (34.6%) |
| BA/BS degree or higher | 88/419 (21.0%) |
| Missing | 3 |
| **Currently a Student** | |
| Yes | 53/419 (12.6%) |
| No | 366/419 (87.4%) |
| Missing | 3 |
| **Employment Status** | |
| Employed full−time | 218/419 (52.0%) |
| Employed part−time | 20/419 (4.8%) |
| Self−employed | 42/419 (10.0%) |
| Unemployed or between jobs | 78/419 (18.6%) |
| On disability | 48/419 (11.5%) |
| Other | 13/419 (3.1%) |
| Missing | 3 |
| **Annual Household Income** | |
| Less than $5,000 | 82/419 (19.6%) |
| $5,000−$29,999 | 124/419 (29.6%) |
| $30,000−$59,999 | 150/419 (35.8%) |
| $60,000 and more | 63/419 (15.0%) |
| Missing | 3 |
| **In the last 6 months how often was there not enough money for rent, food or utilities?** | |
| Never | 159/419 (37.9%) |
| Once in awhile | 136/419 (32.5%) |
| Fairly often | 71/419 (16.9%) |
| Very often | 53/419 (12.6%) |
| Missing | 3 |

*(Continued)*

**Table 1.** (Continued)

| Characteristic | Participants |
|---|---|
| **Health Insurance** | |
| Yes | 270/419 (64.4%) |
| No | 149/419 (35.6%) |
| Missing | 3 |
| **If with health insurance, does it adequately cover your health care needs?** | |
| Yes | 231/270 (85.6%) |
| No | 39/270 (14.4%) |
| **Sexual Orientation** (more than one response could be given) | |
| Homosexual or gay | 299/414 (72.2%) |
| Bisexual | 108/414 (26.1%) |
| Heterosexual or straight | 16/414 (3.9%) |
| Unsure | 5/414 (1.2%) |
| Other | 10/414 (2.4%) |
| Missing | 8 |
| **Married or Living with a Partner** | |
| Yes | 330/420 (78.6%) |
| No | 90/420 (21.4%) |
| Missing | 2 |
| **Living with HIV (self-report)** | |
| Yes | 182/414 (44.0%) |
| No | 204/414 (49.3%) |
| I am not sure | 28/414 (6.8%) |
| Missing | 8 |
| **If not living with HIV or with unknown HIV status, have you been tested for HIV?** | |
| Yes | 213/232 (91.8%) |
| No | 19/232 (8.2%) |

potential participants. Because they had access to so many venues and events, the recruitment teams in these larger communities were able to complete enrollment more quickly (3 months for Dallas/Houston versus 6 months for Montgomery/Greenville). The Greenville recruitment team was also more successful at venue-based recruitment; however, for a very different reason. This recruitment team, which was based out of a community-based organization that provides medical care and supportive services, used their own facility as a venue. To ensure a more representative sample, this team approached every third or fifth potentially eligible Black male (dependent on how busy the schedule was for their facility on any given day) for potential enrollment. The men they approached were coming to their facility for a variety of services (primary care, HIV prevention and testing, mental health services, food pantry, etc.).

In Montgomery, the team had more success with the RDS recruitment component. Due to the size of this study community, coupled with deep-rooted homophobia and racism, this recruitment team relied heavily on RDS because their community landscape didn't have many existing Black MSM-geared venues and events. They created the concept of "RDS parties" that were held at places not known to be affiliated with the gay community, which made some people feel more comfortable attending. They encouraged men enrolled via venue-based recruitment to bring their friends with study coupons to these parties to learn more about the study and, if eligible, enroll on the spot. In addition, the team used social media to encourage those who had received a coupon to attend these parties. This approach allowed them to meet their recruitment goals and tap into a semi-random sample of men in their community, while maintaining the safety and confidentiality of the participants.

For all recruitment teams, venue-based enrollment was capped for each event at 5 enrollments for their first event and 10 enrollments for subsequent events. Towards the end of enrollment, recruitment teams were allowed to enroll up to 12 participants at their last events to complete enrollment, which was approved in advance by the study team.

## HIV status

HIV status was determined by laboratory testing for 403 (95.5%) of the 422 enrolled participants (Table 2). The 19 remaining participants included four with inconclusive HIV test results (reactive Ag/Ab test with an indeterminate Geenius test) and 15 where testing was not performed. The 403 participants with HIV status determined included 212 (52.6%) who were living with HIV and 191 (47.4%) who were not living with HIV. HIV prevalence ranged from 41.8% to 60.0% (Houston 60.0%, Dallas 41.8%, Greenville 49.1%, Montgomery 59.6%).

A higher proportion of men recruited through the RDS component of starfish sampling were living with HIV (61.1%) than among men recruited at venues (50.2%). Among men not living with HIV, PrEP status did not differ significantly by recruitment method; among men living with HIV, rates of viral suppression also did not differ significantly by recruitment method.

## Viral suppression

The proportion of participants with established HIV infections (determined to be living with HIV based on laboratory testing) who were virally suppressed below 200 copies/mL was assessed for the two communities that had plasma stored. The level of viral suppression using this cutoff was 70.3% overall (Houston 81.7%, Dallas 53.7%).

## Acute and early HIV infection

Three (0.7%) of the 403 participants with HIV status determined had HIV RNA detected with an indeterminate Geenius test result (see Laboratory Methodology). Two of these participants reported being in HIV care. One reported taking Biktarvy and had a viral load detected less than the lower limit of detection of the assay (<30 copies/mL). A second participant reported being in HIV care but did not specify a treatment regimen; that participant had no plasma stored for viral load testing. The third participant reported having an HIV test within the prior 3 months and said "no" to the question, "are you living with HIV"; this participant was classified as having an acute HIV infection.

## HIV drug resistance

Twenty-nine samples were tested for drug resistance (19 from Dallas; 10 from Houston); three (10.3%) of the samples had one or more major resistance associated mutation (RAM) detected (Table 3). Two of the three participants were from

**Table 2. HIV Status, PrEP Use and Viral Suppression.**

| | Overall | Houston, TX | Dallas, TX | Greenville, SC | Montgomery, AL |
|---|---|---|---|---|---|
| **Enrolled** | 422 | 105 | 108 | 108 | 101 |
| **Participants with HIV Status Determination** | 403/422 (95.5%) | 100/105 (95.2%) | 98/108 (90.7%) | 106/108 (98.1%) | 99/101 (98.0%) |
| **Not Living with HIV** | 191/403 (47.4%) | 40/100 (40.0%) | 57/98 (58.2%) | 54/106 (50.9%) | 40/99 (40.4%) |
| **Self-reported PrEP Use Among Those Not Living with HIV** | | | | | |
| **I am on PrEP now** | 41/191 (21.5%) | 9/40 (22.5%) | 18/57 (31.6%) | 7/54 (13.0%) | 7/40 (17.5%) |
| **I was on PrEP in the past, but not now** | 16/191 (8.4%) | 6/40 (15.0%) | 5/57 (8.8%) | 3/54 (5.6%) | 2/40 (5.0%) |
| **Never taken PrEP** | 120/191 (62.8%) | 22/40 (55.0%) | 31/57 (54.4%) | 38/54 (70.4%) | 29/40 (72.5%) |
| **Never heard of PrEP** | 8/191 (4.2%) | 2/40 (5.0%) | 1/57 (1.8%) | 4/54 (7.4%) | 1/40 (2.5%) |
| **Living with HIV** | 212/403 (52.6%) | 60/100 (60.0%) | 41/98 (41.8%) | 52/106 (49.1%) | 59/99 (59.6%) |
| **Virally Suppressed** | 71/101 (70.3%) | 49/60 (81.7%) | 22/41 (53.7%) | N/A | N/A |
| **Not Virally Suppressed** | 30/101 (29.7%) | 11/60 (18.3%) | 19/41 (46.3%) | N/A | N/A |

**Table 3. Drug resistance.**

| Site | Viral load (copies/mL) | Aware of HIV status | Self-reported PrEP | Self-reported ART | NRTI RAMs[a] | NNRTI RAMs[a] | PI RAMs[a] | INSTI RAMs[a] | High-level resistance |
|---|---|---|---|---|---|---|---|---|---|
| Dallas | 7,823 | Yes | Descovy (TAF/FTC) | No | | V179I, **G190A** | M50I | | NVP |
| Dallas | 9,641 | Yes | No | Genvoya (EVG/FTC/TFV) | | V108V/I | A71T, **L90M** | | NFV |
| Houston | 300,195 | No | No | No | **M184V**, K70Q | | L10I | **E138K/T**, **S147G**, **R263K**, E157E/Q | BIC, CAB, DTG, EVG, RAL, 3TC, FTC |

[a]Major resistance associated mutations are shown in bold text.

Abbreviations: PrEP: pre-exposure prophylaxis; ART: antiretroviral treatment; NRTI: nucleoside/nucleotide reverse transcriptase inhibitor; NNRTI: non-nucleoside reverse transcriptase inhibitor; PI: protease inhibitor; INSTI: integrase strand transfer inhibitor; RAM: resistance associated mutation; NVP: nevirapine; NFV: nelfinavir; BIC: bictegravir; CAB: cabotegravir; DTG: dolutegravir; EVG: elvitegravir; RAL: raltegravir; 3TC: lamivudine; FTC: emtricitabine.

Dallas, and both reported that they were previously aware they were living with HIV. One participant had the G190A mutation which confers high-level resistance to nevirapine; this participant reported use of emtricitabine/tenofovir alafenamide fumarate (FTC/TAF) for PrEP. The second participant had the L90M mutation which confers high-level resistance to nelfinavir; this participant reported being on ART with elvitegravir, cobicistat, FTC/TAF. The third participant was from Houston. This participant reported no knowledge of a positive HIV status; had the M184V mutation, which confers high-level resistance to lamivudine (3TC) and emtricitabine (FTC); and had the mutations E138K/T, S147G, and R263K, which confers high-level resistance to all integrase strand transfer inhibitors (INSTIs). This participant did not report any use of antiretroviral drugs for PrEP or ART. In each case, the resistance mutations detected were not consistent with self-reported exposure to antiretroviral drugs in the corresponding drug class; this suggests that all three cases may represent cases of transmitted drug resistance, with transmission of multi-class resistant HIV in the third case.

## Treatment cascade

Fig 3 shows the ART treatment cascade. Among those living with HIV overall, 163 (76.9%) reported that they were in care. In addition, 37 participants reported that they were not in care, but were not facing any challenges related to care engagement, and six participants reported they were not in care and did have challenges. Four of these six participants cited the following challenges when seeking care: 1) did not know where to get HIV care, 2) feared being treated poorly by healthcare providers due to their race, 3) did not feel comfortable discussing their sexual activity with their healthcare provider, 4) feared being treated poorly if others knew they were getting tested for HIV, 5) lack of social support, and 6) had alcohol or drugs affect their ability to seek HIV care. Viral load data were only available in Dallas and Houston; however, of those living with HIV in these two communities, 71 (70.3%) were virally suppressed.

## PrEP use

Overall, 188 DBS samples were tested for intraerythrocytic TFV-DP, with 36 samples having quantifiable TFV-DP concentrations (above 31.3 fmol/punch[es]). Forty-one participants reported currently taking PrEP. This included 15 participants who reported taking Truvada, Tenvir-EM or an unknown oral PrEP regimen, of which 11 (73.3%) had a quantifiable concentration of TFV-DP in their DBS sample. Based on the concentrations of TFV-DP measured, on average, four were taking fewer than two doses per week, one was taking two to three doses per week, and six were taking four or more doses per week. Twenty-six participants reported taking Descovy, of which 22 (84.6%) had a quantifiable concentration of TFV-DP in their DBS sample. Based on the concentrations of TFV-DP measured, on average, five were taking fewer than two doses per week, three were taking two to three doses per week, and 14 were taking four or more doses per week.

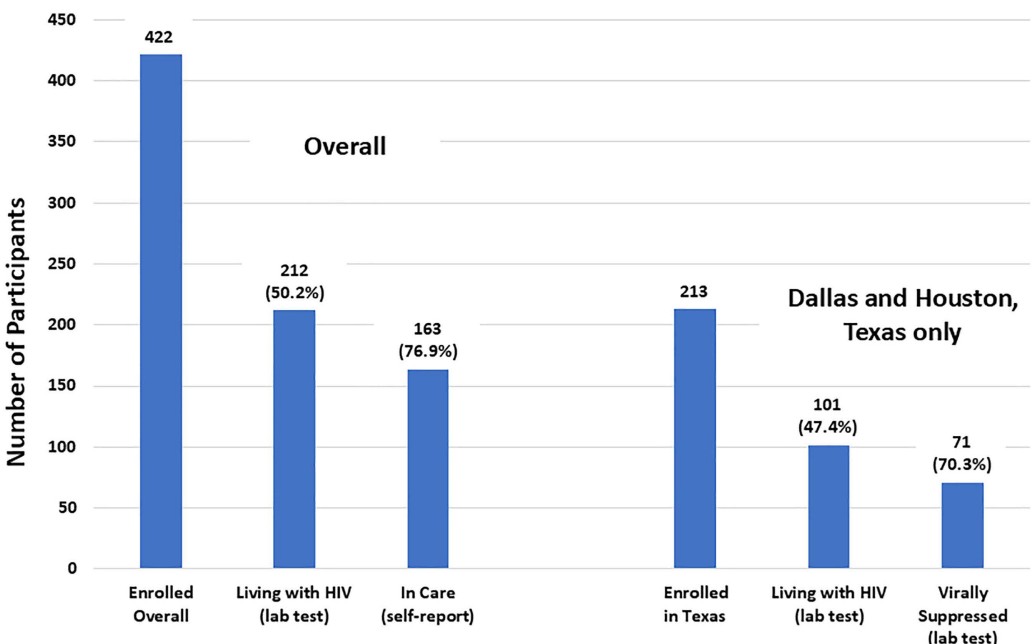

**Fig 3. Treatment cascade.**

## PrEP cascade

Fig 4 depicts the PrEP cascade. Out of those not living with HIV, 57 (29.8%) reported ever taking PrEP and 41 (21.5%) reported being currently on PrEP. Of the 41 reporting current PrEP use, 33 (80.5%) had a quantifiable concentration of TFV-DP in their DBS sample. In addition, 120 (62.8%) reported never taking PrEP and 8 (4.2%) reported never having

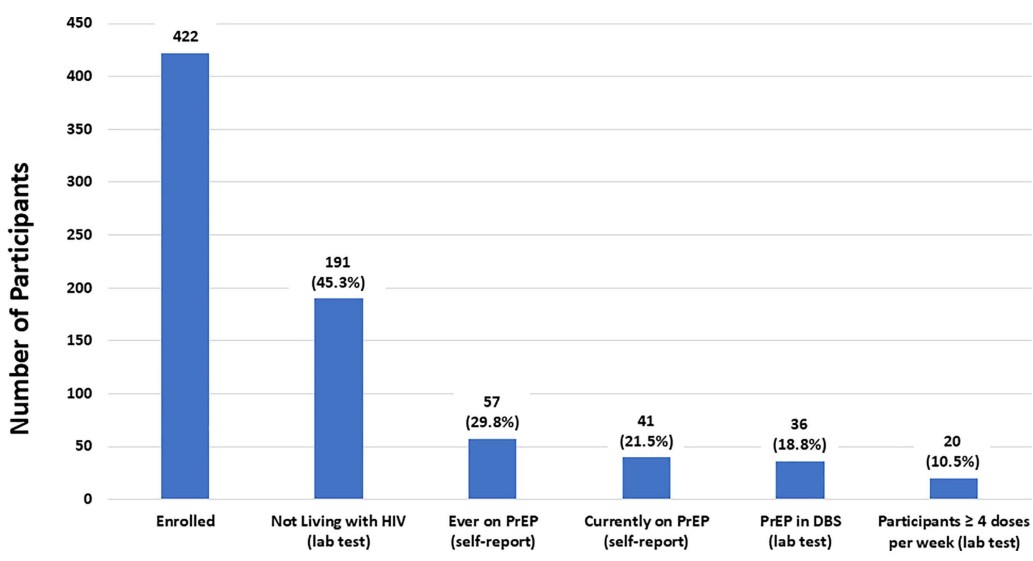

**Fig 4. PrEP cascade.**

heard of PrEP. One hundred eighteen participants not living with HIV reported that they were not on PrEP and were not facing any challenges in accessing or taking it. However, for those who reported that they were not on PrEP and had faced challenges (13 participants), the reasons cited most were paying for medication (9 participants), medical visits or lab tests (8 participants), and taking a pill every day (7 participants).

## Discussion

The HPTN 096 cross-sectional assessment is the first HPTN study to use starfish sampling. This method proved to be efficient and effective for timely recruitment of Black MSM in each city. The flexibility of starfish sampling allowed for successful recruitment in both larger and smaller cities, including those like Greenville and Montgomery, that have few, or no, venues or events dedicated to Black MSM. Using this approach, we found a very high burden of HIV; in two communities, more than half of the men were living with HIV. This high HIV prevalence is consistent with 2017 CDC estimates that the lifetime acquisition probability for HIV among Black gay and bisexual men in the US is 50% [13]. Another explanation for the high HIV prevalence in this group may be that the venue time component of starfish sampling could have been biased in favor of recruiting gay-identified men who may have higher burdens of HIV acquisition than men who may not identify as gay or men who do not routinely attend gay-friendly venues. The same potential bias may help explain the relatively higher rate of PrEP awareness among the men not living with HIV (96%), compared to national estimates for Black MSM (90%) [14], which may be reflective of a more engaged and health literate group of men attending gay-friendly venues. Despite the high level of PrEP awareness in the sample, the actual rate of PrEP use (30%) was somewhat lower than national estimates for Black MSM (38%) [14], perhaps because the assessment was carried out in study communities located in the Southern US.

We did identify discrepancies between self-reported PrEP use and PrEP as measured in DBS. These may be explained by a proportion of men reporting PrEP use, and using intermittent (or on-demand) regimens as opposed to daily dosing. On-demand dosing is commonly used by men with more episodic risks and may lead to negative DBS findings if men have not recently used PrEP.

It is well established that persons living with HIV who are stably virally suppressed do not transmit HIV to their sexual partners [15–17]; for this reason, rates of viral suppression in networks and populations almost certainly impact HIV incidence at population levels. We found an overall prevalence of viral suppression of 70% in the cities where viral load data were available (Dallas and Houston). This is in line with the national average for viral suppression in 2022 among Black MSM (61.8%), all MSM (67.9%) and for all Americans (65.1%) in the US [18] and is clear evidence of an unmet need for enhanced treatment efforts for these men. Our inability to assess viral suppression for logistical reasons in Greenville and Montgomery is a study limitation.

The cross-sectional assessment helped inform the sample size for the larger efficacy trial. We expected an HIV prevalence of 20–30%. Our finding that the majority of men enrolled were living with HIV meant that we would need to increase the sample size significantly to ensure recruitment of enough men not living with HIV. For reasons unrelated to the outcome of the cross-sectional assessment, the larger HPTN 096 trial will not go forward with the original study design. Even so, the cross-sectional assessment provides key data needed to inform the EHE initiative [7] and other pivotal HIV research.

As noted above, assessments for viral load and HIV drug resistance could only be performed for two of the four communities (Dallas and Houston). The frequency of drug resistance in this group (3 of 29 cases assessed, 10.3%) is lower than the frequency observed in other studies of MSM in the US. In a study of Black MSM in the US enrolled in 2009–2010, 28% of participants had resistance to non-INSTI drugs [19] with no INSTI resistance detected [20]. In a study that enrolled MSM in four US cities in 2016–2017 (68% Black), INSTI resistance was observed in 8% of participants; overall rates of drug resistance ranged from 21% to 53% across study sites and 12% of the men had multiclass resistance [21]. The lower frequency of resistance to non-INSTI drugs in this cross-sectional assessment most likely reflects the decline in

use of those drugs for ART; however, it could also reflect differences in the location of study communities, methods used for participant recruitment, or other factors. The low rate of INSTI resistance in this group (one of 29 cases assessed, 3.4%) is encouraging for use of long-acting cabotegravir PrEP and INSTI-based ART in this population. Detection of three cases with possible transmission of resistant HIV (including resistance to oral PrEP medications, INSTIs, and multiple drug classes) is concerning. However, it is possible that resistance was acquired in one of more of these cases since prior studies have shown that some participants in research studies may chose not to report knowledge of their HIV status or ART history [22].

## Limitations

This pilot study had several limitations, the primary one being that the Montgomery and Greenville communities did not have an LDMS laboratory that was close enough to allow sample transport, processing, and plasma storage within the required time windows for viral load and HIV drug testing. This prevented the team from determining the prevalence in these two communities. In addition, 78% of the participants were recruited via venue-based sampling, which may have overrepresented Black MSM who are open about their sexual orientation. Black MSM who attend gay-friendly venues may have different risk factors for HIV acquisition compared to those who are less open and more discreet about their sexual orientation.

## Conclusions

The HPTN 096 pilot cross-sectional assessment provided key information about the Black MSM population in four cities in the Southern US. Starfish sampling proved an important new recruitment tool for this population and should be further investigated for other populations and settings, as well as for measures of generalizability and representativeness. The HIV epidemic among Black MSM in the Southern US is a critical public health emergency and must be addressed with all available approaches. More than 13 years into the PrEP era, and 30 into the ART era, we can and must do better for these men.

## Supporting information

**S1 Table.  HPTN 096 Study Community Pairs and Randomization.**
(DOCX)

## Acknowledgments

We thank Willie McFarlen and Henry Fisher Raymond for their help with the starfish sampling methodology. We also thank Michael Stirratt for continued support and guidance for study design and implementation. Finally, we thank Dawn Smith (1949–2022) for her support and data provision for the study, as well as her significant contributions to the fight against HIV, particularly in her work with underrepresented communities. She will be remembered as a respected and well-loved colleague, a mentor to many fellows and younger staff, and a shining example of scientific integrity, intellectual rigor, and dedication to social justice and public health.

    The members of the HPTN 096 pilot study team include: LaRon E. Nelson, Chris Beyrer, Robert H. Remien, Dale Burwen, Michael J. Stirratt, Kevin P. Delaney, Anna Satcher Johnson, Dawn K. Smith, Kirsten Argueta, Stacy Cohen, Shelita Merchant, Kevin Bokoch, Marcus D. Bryan, Theresa R. Gamble, Elizabeth Greene, Sherri Johnson, Rita Labbett, Laura Long, Jonathan Paul Lucas, Kathleen M. MacQueen, Daniel D. Driffin, Christopher Hucks-Ortiz, Craig S. Hutchinson, Melissa M. Turner, Lynda Marie Emel, Tanya Harrell, James P. Hughes, Allison Meisner, Julie Ngo, Susan H. Eshleman, Philip A. Sullivan, Charlotte A. Gaydos, Christoph C. Carter, James F. Rooney, Melverta Bender, Melvin L. Cauthen, Kendrick Clack, Rochelle Cole, Matthew Jenkins, Venton C. Hill-Jones, Daniel Murdock, Ashley Ross, Justin C. Smith,

Bruce Agins, Marlon M. Bailey, Donte T. Boyd, Donaldson Conserve, Patricia Coury-Doniger, Stephan Davis, Dustin T. Duncan, Errol L. Fields, Sheldon D. Fields, Lisa Hightow-Weidman, Mandy J. Hill, Risha Irvin, Alex S. Keuroghlian, Tavell L. Kindall, Carmen Logie, Kenneth H. Mayer, Leisha McKinley-Beach, Leandro A. Mena, Ofole Mgbako, Kate M. Mitchell, Steve North, Gjvar M. Payne, Julie Pulerwitz, Patrick Sullivan, DeAnne Turner, David Vlahov, Stephaun Wallace, Darren L. Whitfield.

**Disclaimer**: The content of this article is solely the responsibility of the authors and does not necessarily represent the official views of the National Institutes of Health. The mention of any specific commercial products, process, service, manufacturer, company or organization does not constitute its endorsement or recommendation by the U.S. Government or the National Institutes of Health.

## Author contributions

**Conceptualization:** Chris Beyrer, Robert H. Remien, Susan H. Eshleman, Theresa R. Gamble, Rita L. Labbett, James P Hughes, Daniel D. Driffin, Craig S. Hutchinson, Christopher Hucks-Ortiz, Donte Boyd, Dale R. Burwen, Anna Satcher Johnson, LaRon E. Nelson.

**Data curation:** Theresa R. Gamble, Jean De Dieu Tapsoba, Philip A. Sullivan, Oliver Laeyendecker, Peter L. Anderson, Devang Agravat, Anna Satcher Johnson.

**Formal analysis:** Susan H. Eshleman, Jean De Dieu Tapsoba, Oliver Laeyendecker, Peter L. Anderson, Devang Agravat.

**Funding acquisition:** Chris Beyrer, Theresa R. Gamble, Dale R. Burwen, LaRon E. Nelson.

**Investigation:** Oliver Laeyendecker, Peter L. Anderson, Maurice Adair, Melissa Curry, Shakita Brooks Jones, Ian L. Haddock.

**Methodology:** Chris Beyrer, Susan H. Eshleman, Oliver Laeyendecker, Peter L. Anderson.

**Project administration:** Theresa R. Gamble, Rita L. Labbett, Philip A. Sullivan.

**Supervision:** Chris Beyrer, Susan H. Eshleman, Theresa R. Gamble, Rita L. Labbett, Philip A. Sullivan, James P Hughes, Maurice Adair, Melissa Curry, Shakita Brooks Jones, Ian L. Haddock, Dale R. Burwen, LaRon E. Nelson.

**Writing – original draft:** Chris Beyrer, Susan H. Eshleman, Theresa R. Gamble, Jean De Dieu Tapsoba, Rita L. Labbett, Peter L. Anderson.

**Writing – review & editing:** Chris Beyrer, Robert H. Remien, Susan H. Eshleman, Theresa R. Gamble, Jean De Dieu Tapsoba, Rita L. Labbett, Philip A. Sullivan, Oliver Laeyendecker, Peter L. Anderson, Devang Agravat, James P Hughes, Daniel D. Driffin, Craig S. Hutchinson, Christopher Hucks-Ortiz, Maurice Adair, Melissa Curry, Shakita Brooks Jones, Ian L. Haddock, Donte Boyd, Dale R. Burwen, Anna Satcher Johnson, LaRon E. Nelson.

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
