## [Decision Letter · Decision Letter 0]

21 Jul 2025

Dear Dr. Beyrer,

Thank you for submitting your manuscript to PLOS ONE. After careful consideration, we feel that it has merit but does not fully meet PLOS ONE’s publication criteria as it currently stands. Therefore, we invite you to submit a revised version of the manuscript that addresses the points raised during the review process.

We look forward to receiving your revised manuscript.

Kind regards,

Douglas S. Krakower, MD

Academic Editor

PLOS ONE

Journal Requirements:

[Overall support for the HIV Prevention Trials Network (HPTN) is provided by the National Institute of Allergy and Infectious Diseases (NIAID), Office of the Director (OD), National Institutes of Health (NIH), National Institute on Drug Abuse (NIDA), the National Institute of Mental Health (NIMH), and the Eunice Kennedy Shriver National Institute of Child Health and Human Development (NICHD) under Award Numbers UM1AI068619 (HPTN Leadership and Operations Center), UM1AI068617 (HPTN Statistical and Data Management Center), and UM1AI068613 (HPTN Laboratory Center].

3. Thank you for stating the following in your manuscript:

[Overall support for the HIV Prevention Trials Network (HPTN) is provided by the National Institute of Allergy and Infectious Diseases (NIAID), Office of the Director (OD), National Institutes of Health (NIH), National Institute on Drug Abuse (NIDA), the National Institute of Mental Health (NIMH), and the Eunice Kennedy Shriver National Institute of Child Health and Human Development (NICHD) under Award Numbers UM1AI068619 (HPTN Leadership and Operations Center), UM1AI068617 (HPTN Statistical and Data Management Center), and UM1AI068613 (HPTN Laboratory Center).]

[Overall support for the HIV Prevention Trials Network (HPTN) is provided by the National Institute of Allergy and Infectious Diseases (NIAID), Office of the Director (OD), National Institutes of Health (NIH), National Institute on Drug Abuse (NIDA), the National Institute of Mental Health (NIMH), and the Eunice Kennedy Shriver National Institute of Child Health and Human Development (NICHD) under Award Numbers UM1AI068619 (HPTN Leadership and Operations Center), UM1AI068617 (HPTN Statistical and Data Management Center), and UM1AI068613 (HPTN Laboratory Center]

[The authors have declared that no competing interests exist.].   

We note that one or more of the authors are employed by a commercial company: D3 Consulting LLC

6. We note that Figure 4 in the Supplementary File of your submission contains map image which may be copyrighted. All PLOS content is published under the Creative Commons Attribution License (CC BY 4.0), which means that the manuscript, images, and Supporting Information files will be freely available online, and any third party is permitted to access, download, copy, distribute, and use these materials in any way, even commercially, with proper attribution. For these reasons, we cannot publish previously copyrighted maps or satellite images created using proprietary data, such as Google software (Google Maps, Street View, and Earth). For more information, see our copyright guidelines: http://journals.plos.org/plosone/s/licenses-and-copyright.

1. You may seek permission from the original copyright holder of Figure 4 in your Supplementary File to publish the content specifically under the CC BY 4.0 license. 

7. We notice that your supplementary tables are included in the manuscript file. Please remove them and upload them with the file type 'Supporting Information'. Please ensure that each Supporting Information file has a legend listed in the manuscript after the references list.

Reviewers' comments:

Reviewer's Responses to Questions

**Comments to the Author**

1. Is the manuscript technically sound, and do the data support the conclusions?

Reviewer #1: Yes

Reviewer #2: Yes

2. Has the statistical analysis been performed appropriately and rigorously?

Reviewer #1: Yes

Reviewer #2: Yes

3. Have the authors made all data underlying the findings in their manuscript fully available?

Reviewer #1: Yes

Reviewer #2: Yes

4. Is the manuscript presented in an intelligible fashion and written in standard English?

Reviewer #1: Yes

Reviewer #2: Yes

Reviewer #1: In this manuscript the Authors describe their findings from a cross-sectional assessment during the pilot phase of the HPTN 096 study. Overall, they found a very high burden of HIV in Black MSM at the study sites. Of those living with HIV, 77% were in HIV care. At two study sites, viral loads were obtained and only 70% were virally suppressed. Of those not living with HIV, only 22% reported being on PrEP, and of those, 12% had no evidence of PrEP on dried blood spots. Overall, the study Authors found a high burden of HIV in Black MSM and low rates of viral suppression and PrEP use in those not living with HIV.

I found this manuscript interesting and well-written and only have some minor suggestions:

1. "southern" should be capitalized throughout the manuscript when referring to "Southern United States".

2. In the Introduction, both present and past tense are used. I would stick with past tense whenever applicable.

3. Line 251-253: This is in track change.

4. Line 276: This is in track change.

5. Line 285-286: What do you mean by "HIV testing was performed for the remaining four participants..." Please consider rephrasing.

6. Lines 411-414: Of those reporting taking PrEP but without detectable TFV-DP on their DBS, were you able to confirm that they were not on injectable PrEP (Cabotegravir was approved by the FDA for PrEP in 2021 before this study)?

Reviewer #2: This is a remarkably well-written and comprehensive description of the cross-sectional study data from HPTN 096. I only have two minor points of clarification.

Were any efforts made to correct for the RDS component of the sampling strategy in the analysis?

Only 29 samples were tested for HIV drug resistance. A little more information about this and if there is any suspected bias in which participants had samples tested for resistance compared to those who did not would be helpful. Was this just due to the differences in lab accessibility across communities? It seems as though there are still missing samples even with this limitation.

Although not pertinent to the review, I found the tribute to Dawn Smith to be quite moving. Having known her and the standards she set for herself and those she worked with, it is fitting for this acknowledgment to be attached to such a thorough, clearly presented, and important paper.

**Do you want your identity to be public for this peer review?** For information about this choice, including consent withdrawal, please see our Privacy Policy

Reviewer #1: **Yes: ** Jeffrey D. Jenks

Reviewer #2: No

---

## [Author Response · Author response to Decision Letter 1]

16 Sep 2025

Response to Reviewers included in Response to Reviewers letter.

The manuscript with recent tracked changes has been uploaded including the correction in affiliations.

---

## [Editor Report · Decision Letter 1]

22 Sep 2025

Investigating the HIV epidemic among Black gay and bisexual men in the southern United States: Results of the HPTN 096 pilot cross-sectional assessment

PONE-D-25-32729R1

Dear Dr. Beyrer,

We’re pleased to inform you that your manuscript has been judged scientifically suitable for publication and will be formally accepted for publication once it meets all outstanding technical requirements.

Kind regards,

Douglas S. Krakower, MD

Academic Editor

PLOS ONE
---

## [Editor Report · Acceptance letter]

PONE-D-25-32729R1

PLOS ONE

Dear Dr. Beyrer,

I'm pleased to inform you that your manuscript has been deemed suitable for publication in PLOS ONE. Congratulations! Your manuscript is now being handed over to our production team.

Kind regards,

on behalf of

Dr. Douglas S. Krakower

Academic Editor

PLOS ONE